# Effects of Heat Treatment Atmosphere and Temperature on the Properties of Carbon Fibers

**DOI:** 10.3390/polym14122412

**Published:** 2022-06-14

**Authors:** Gyungha Kim, Hyunkyung Lee, Kyungeun Kim, Dae Up Kim

**Affiliations:** Carbon & Light Materials Application Group, Korea Institute of Industrial Technology, 222, Palbok-ro, Deokjin-gu, Jeonju-city 54853, Jeollabuk-do, Korea; gyungha@kitech.re.kr (G.K.); dori9424@kitech.re.kr (H.L.); kke@kitech.re.kr (K.K.)

**Keywords:** carbon fiber, heat treatment, functional group, surface free energy, mechanism

## Abstract

In this study, carbon fibers were heat-treated in a nitrogen and oxygen atmosphere according to temperature to elucidate the mechanism of chemical state changes and oxygen functional group changes on the carbon fiber surface by analyzing the mechanical and chemical properties of carbon fibers. Carbon fibers before and after heat treatment were analyzed using FE-SEM (Field Emission Scanning), UTM (Universal Tensile Testers), XPS (X-ray Photoelectron Spectroscopy), and surface-free energy. In the nitrogen atmosphere, which is an inert gas, the tensile strength was equivalent to that of the virgin up to 500 °C but decreased to 71% with respect to the virgin at 1000 °C. Furthermore, as the temperature increased from room temperature to 1000 °C, the oxygen functional group and the polar free energy gradually decreased compared with the virgin. On the other hand, in the oxygen atmosphere, which is an active gas, the tensile properties were not significantly different from those of the virgin up to 300 °C but gradually decreased at 500 °C. Above 600 °C, the carbon fibers deteriorated, and measurement was impossible. The oxygen functional group decreased at 300 °C, but above 300 °C, among the oxygen functional groups, the hydroxyl group and the carbonyl group increased. Furthermore, the lactone group formed and rapidly increased compared with the virgin, and the polar free energy increased as the temperature increased.

## 1. Introduction

In recent years, carbon fiber composites have expanded their application not only in the aerospace industry but also in all industrial fields due to their high tensile strength and rigidity, excellent thermal conductivity, and electrical conductivity [1,2,3,4,5,6,7,8]. However, despite these advantages, carbon fiber composites are applied to produce only expensive parts due to their high price [9], and they have problems of environmental pollution since they are mainly manufactured from thermosetting resin-based composite materials that are difficult to recycle. These limitations point to the need for research on recycling carbon fiber composites to secure price competitiveness and solve environmental problems. Currently, a method for recovering recycled carbon fibers has been developed, but the recovered recycled carbon fibers have deteriorated mechanical properties compared with the virgin, and there is a limit to reusing them [10,11,12]. In particular, the properties of carbon fibers are greatly affected by the interfacial properties of the surface, and if the interfacial properties of carbon fibers and resin are low, the mechanical properties of the carbon composite deteriorate. Therefore, research and development to improve interfacial properties through various surface treatments are underway [13,14,15,16,17].

To improve the properties of recycled carbon fibers to the same level as the virgin, a surface treatment is employed, which binds polarities such as oxygen functional groups to the surface, thus improving the interfacial properties of the recovered carbon fibers. Surface treatment is an essential process, especially for non-polar resin polypropylene [14].

In general, the surface treatments of carbon fibers include a wet method (liquid oxidation) [16,18,19,20,21,22] in which the surface is treated with an acid such as sulfuric acid or nitric acid, a dry method (dry gaseous oxidation) in which the surface is treated at a high temperature in an oxidizing gas or inert gas atmosphere [9,23,24,25,26,27], electrochemical oxidation [4,7,28], a plasma method of surface treatment using ionized gas [29,30,31], or energetic ion oxidation that exposes them to strong energy such as ozone and ultraviolet rays [3,32,33,34]. It is known that heat treatment, which is a dry method in the surface treatment methods of carbon fibers, changes the properties of the carbon fibers according to the change in the atmosphere, temperature, and treatment time [23,24]. Song observed a resin that could not be decomposed by heat treatment on the surface of carbon fiber when the carbon composite was heat-treated in a nitrogen atmosphere at a temperature of 500 °C [9]. Liansheng et al. confirmed that when carbon fibers were heat-treated at 1500 °C in a nitrogen atmosphere, the degree of damage to carbon fibers increased with the temperature, and tensile strength and fracture toughness decreased [27]. In addition, as a result of analyzing the physical and chemical properties of carbon fibers according to the temperature of the heat treatment in a vacuum atmosphere, it was reported that as the temperature of the heat treatment increased, the contact angle increased, and the polar surface free energy and IFSS (Interface shear strength) decreased due to the decrease in the carbon fiber hydroxyl group [25]. On the other hand, as a result of heat treatment, by changing the amount of H_2_ while keeping Ar constant, the change in the components of carbon, oxygen, and nitrogen was small regardless of the amount of H_2_, but the surface roughness and the interfacial properties increased as the amount of H_2_ increased [26]. As a result of heat treatment in an O_2_/(O_2_ + N) atmosphere and analyzing changes in the chemical properties of the carbon fiber surface using XPS (X-ray photoelectron spectroscopy), oxygen existing in the atmosphere was introduced to the carbon fiber surface, and the amount of carboxyl group and ILSS (Interlaminar Shear Strength) increased compared to that of the virgin [24]. According to the research so far, the optimum conditions depending on the atmosphere of the heat treatment and process conditions are shown, but the claims about the change in the properties of carbon fibers due to the atmosphere of the heat treatment are different, and the main causes are insufficient.

In this study, a heat treatment process was introduced to improve the properties of recycled carbon fibers, and optimum conditions for ensuring that the interface properties between carbon fiber and resin were improved through mechanical and chemical property analysis by heat treatment temperature in an oxygen and nitrogen atmosphere. Furthermore, the mechanism for the change in the chemical state of the carbon fiber surface and the change in the oxygen functional group due to the atmosphere and the temperature of the heat treatment was determined.

## 2. Experimental Details

The carbon fiber used in this study was T700SC manufactured by Toray, Japan, which has an average diameter of approximately 7 μm and is based on PAN. PAN-based carbon fiber is made by a method of stabilizing the PAN-based fiber by treating it with flame resistance and then performing carbonization firing or graphitization in an inert atmosphere [35]. The carbon fibers used in the experiment were downsized and then heat-treated in a tube furnace. The heat treatment was performed in an oxygen and nitrogen atmosphere at a temperature of 300 to 1000 °C, a time of 1 h, a heating rate of 5 °C/min, and a flow rate of 2 cc/min. 

To observe the surface changes of the virgin and the heat-treated carbon fibers, the observation was performed using FE-SEM under the conditions of an acceleration voltage of 20 kV. To evaluate the mechanical properties of carbon fibers, a short fiber tensile test was performed under the conditions of a tensile speed of 5 mm/min according to the ASTM D3822 standard. The test was performed 25 times or more per test condition, and the average value was used. XPS from the Nexsa XPS system (Korea Basic Science Institute, Jeonju, Korea) was used to analyze changes in the chemical function of the carbon fiber surface due to heat treatment conditions. The test piece was irradiated with a monochromatic Al Kα source (1486.6 eV), and the high-resolution spectrum imaged a beam with a passing energy of 50 eV and a magnitude of 400 μm. Dynamic contact angle measurement is required to analyze surface energy changes, and in this study, it was measured by the Wilhelmy plate method using hydrophilic water and hydrophobic diiodomethane for dynamic contact angle measurement. The dynamic contact angle and surface free energy were calculated from the values.

## 3. Results and Discussion

### 3.1. Surface Topography of the Carbon Fibers

Figure 1 shows the observation results of the surface of carbon fibers according to the atmosphere and temperature of the heat treatment. When the heat treatment was performed in a nitrogen atmosphere, no change in the carbon fiber surface could be observed up to 500 °C. However, as the temperature of the heat treatment increased, at a temperature of 1000 °C, as shown in Figure 1d, the surface of the fiber was damaged, and deeply dug grooves were observed. When the heat treatment was performed in an oxygen atmosphere, the condition of the surface did not change significantly up to 300 °C, but the carbon fibers reacted with the oxygen atmosphere of heat treatment at 500 °C. In an oxygen atmosphere, defects other than those found in a nitrogen atmosphere were observed. At 600 °C, since the reaction between the carbon fiber and the oxygen in the oxygen atmosphere was intense, the diameter of the carbon fiber was significantly reduced, or a part of the carbon fiber disappeared.

According to other researchers, as a result of heat treatment in the H_2_/Ar atmosphere, which is an inert gas, there is almost no surface reaction at 600 to 700 °C [26], and when heat treatment is performed in a nitrogen atmosphere, defects exist on the carbon fiber surface above 600 °C [9]. It can be seen that the surface reaction and deterioration of the carbon fibers differ greatly depending on the atmosphere during the heat treatment, and the atmosphere of the heat treatment has a greater effect than the temperature of the heat treatment. This means that when heat treatment is performed in an oxygen atmosphere, which is an active gas, more oxygen permeates the carbon fibers than in a nitrogen atmosphere, which is an inert gas, and the bonds between carbons that exist inside the carbon fibers are easily decomposed. As a result of this, the carbon fibers are damaged.

### 3.2. Tensile Properties of Carbon Fibers

The tensile properties according to the temperature of the heat treatment in the oxygen and nitrogen atmosphere were evaluated, and the results are shown in Figure 2. The tensile strength and elongation after heat treatment showed similar properties. That is, it was almost the same as that of the virgin up to 500 °C in a nitrogen atmosphere but decreased sharply to 1.26 GPa at 1000 °C. In the oxygen atmosphere, there was no big difference from that of the virgin up to 300 °C, but it gradually decreased after 400 °C, and at 600 °C, it deteriorated where it was impossible to evaluate the tensile properties. In the nitrogen atmosphere, the modulus was similar to that of the virgin up to 500 °C and decreased by 10% at 1000 °C compared to that of the virgin. However, in an oxygen atmosphere, the modulus was slightly reduced at 500 °C compared to that of the virgin, but analysis was not possible at 600 °C.

According to the study by Song, it was confirmed that the tensile strength was slightly reduced when heat-treated after 600 °C in a nitrogen atmosphere [9]. In the study by Rong et al., when heat-treated in an oxygen atmosphere, pitting occurred with an increase in the surface area on the fiber surface without a change in tensile strength up to 1 h at 400 °C, but the tensile strength decreased as pitting increased and the surface area gradually decreased from 2 h [36]. Moreover, in this study, when the temperature of the heat treatment was 500 °C, it was confirmed that the difference in surface damage and tensile properties due to the atmosphere of the heat treatment was larger than the effect of temperature. In general, when carbon fiber is heat-treated, it is believed that the number and size of pores existing on the surface increase, which weakens the mechanical properties of carbon fiber. Furthermore, it is observed that the nitrogen atmosphere, which is an inert gas, reacts only on the surface of the carbon fiber, but the oxygen atmosphere, which is an active gas, reacts violently not only with the surface but also with the carbon inside the carbon fiber after 300 °C, causing deterioration of the carbon fiber and the loss of mechanical strength.

### 3.3. Surface Composition of Carbon Fibers

Figure 3 shows the XPS spectra of carbon fibers in order to analyze the chemical changes on the surface of the carbon fibers according to the conditions of the heat treatment. The composition changes and the O/C ratios are summarized in Table 1. After heat treatment in a nitrogen atmosphere, as the temperature increased to 500 °C, the amount of carbon relative to that of the virgin increased, the amount of oxygen decreased, and the amount of nitrogen and silicon increased slightly. On the other hand, observing the changes in the components before and after the heat treatment in the oxygen atmosphere, the amount of carbon increased at 300 °C and decreased at 500 °C with respect to the virgin, and the amount of oxygen showed a tendency opposite to the amount of carbon. After heat treatment, the amounts of nitrogen and silicon increased compared to those of the virgin and increased more in the oxygen atmosphere than in the nitrogen atmosphere. Looking at the O/C ratio [18], which is an index of interfacial properties, it decreased to 0.14 compared to that of the virgin at 500 °C when heat-treated in a nitrogen atmosphere. However, after heat treatment in an oxygen atmosphere, it decreased at 300 °C compared to that of the virgin and increased significantly to 0.39 at 500 °C.

Looking at the C1s spectrum, after heat treatment in a nitrogen atmosphere, the hydroxyl group (C–O) and carbonyl group (C=O), which were abundantly present on the carbon fiber surface, were significantly reduced at 300 °C compared to those of the virgin, and at 500 °C compared to 300 °C, the amount of C–O and C=O decreased slightly (Figure 3a,b). In the oxygen atmosphere, the amount of C–O and C=O decreased significantly at 300 °C compared to that of the virgin, but the amount of C–O and C=O increased at 500 °C compared to 300 °C, and a new lactone group (O=C–O) formed and also increased sharply (Figure 3c,d). This is because when heat treatment was performed in a nitrogen atmosphere, oxygen atoms of C–O and C=O present on the carbon fiber surface were removed to O_2_ by heat, and the oxygen functional group was gradually reduced compared to the virgin. On the other hand, in an oxygen atmosphere of up to 300 °C, the oxygen atoms of C–O and C=O of carbon fibers and the atoms in the oxygen atmosphere are removed by O_2_ and heat, and the oxygen functional groups slightly decrease. This is similar to the stabilization process, which is a pretreatment process during the carbonization process [37,38]. After 500 °C, it reacts violently with the carbon present inside the carbon fiber and is removed by CO and CO_2_, and the carbon inside the carbon fiber decreases [39]. The atoms in the oxygen atmosphere penetrate the surface of the carbon fiber. C–O and C=O increase, and O=C–O rapidly forms and increases, resulting in a significant increase in oxygen functionality.

The separations of C1s are shown in Figure 4 and Table 2 to investigate the change of the peak according to the composition of functional groups formed on the carbon fiber surface. As can be seen from the above results, when the temperature increased from 300 °C to 500 °C after heat treatment in a nitrogen atmosphere, C–C, C=C, and C–N increased compared to those of the virgin, C–O decreased significantly, and O=C–O slightly increased. After heat treatment in an oxygen atmosphere, C=Csp2 and C–N increased, C–O decreased significantly, and O=C–O increased slightly at 300 °C compared to the virgin, but at 500 °C compared to 300 °C, C=Csp2 decreased significantly, C–N and C–O increased, and O=C–O increased significantly. From this result, it was found that the oxygen functional group was removed by a large decrease in C–O and C=O as the temperature increased to 1000 °C after heat treatment in a nitrogen atmosphere. However, in an oxygen atmosphere, it was determined that the oxygen functional group greatly increased with the increase in C–O and O=C–O after 300 °C.

Previous studies have reported that during heat treatment in an oxygen atmosphere, oxygen breaks the N–H bound to the carbon fiber surface and binds to oxygen to increase the amount of N–O, C–O, and C=O [24]. In addition, studies on heat treatment in the H_2_/Ar atmosphere reported that there was a slight surface change due to hydrogen, but there was almost no chemical change [26]. After surface treatment with nitric acid, nitric acid has been shown to break C=C bonds and increase oxygen functional group bonds [16]; after plasma treatment in an oxygen atmosphere, the carbonyl group, carboxyl group, and ester group increased the oxygen on the surface of the carbon fiber [29]. After anodic oxidation treatment using H_3_PO_4_, when the current density increases, oxygen functional groups are introduced on the surface of the carbon fiber up to a certain current density, and C–O, C=O, and O=C–O increase [28]. In this study, as the temperature of the heat treatment increases in a nitrogen atmosphere, the oxygen functional group gradually decreases, and in the oxygen atmosphere, when the temperature rises above 500 °C, the oxygen functional group increases due to the formation and rapid increase in O=C–O. From this result, it is inferred that the interfacial properties inside the carbon fiber will increase.

### 3.4. Surface Energy Analysis

To consider the change in the surface free energy of carbon fibers after heat treatment, the dynamic contact angle was measured using a hydrophilic wet liquid and a hydrophobic wet liquid, and the dynamic contact angle was substituted into Equation (1) [40]. Then, the polar surface free energy and nonpolar surface free energy values were calculated.
(1)$$\frac{\gamma_{L}(1+\cos\theta )}{2{\left(\gamma_{L}^{D}\right)}^{\frac{1}{2}}}={\left(\gamma_{S}^{P}\right)}^{\frac{1}{2}}\times {\left(\frac{\gamma_{L}^{P}}{\gamma_{L}^{D}}\right)}^{\frac{1}{2}}+{\left(\gamma_{S}^{D}\right)}^{\frac{1}{2}}$$

The polar surface energy and the nonpolar surface energy were obtained by substituting the dynamic contact angle created when the sample entered the hydrophilic and hydrophobic wet liquid into Equation (1). The dynamic contact angle of the carbon fiber according to the conditions of the heat treatment is shown in Figure 5. The dynamic contact angle was hardly affected by the atmosphere of the heat treatment at 300 °C, but after that, it increased slightly in the nitrogen atmosphere compared to the virgin and decreased significantly in the oxygen atmosphere as compared with the nitrogen atmosphere.

Figure 6 shows the results of classification according to the atmosphere and temperature of the heat treatment into polar free energy and nonpolar free energy from the dynamic contact angle results. In the nitrogen atmosphere, there was almost no change in the surface free energy up to a temperature of 500 °C, and the polar free energy decreased slightly as the temperature increased. On the other hand, in the oxygen atmosphere, the surface energy gradually increased as the temperature of the heat treatment increased, and the ratio of the polar free energy tended to gradually increase.

Viewing the polarity/surface free energy ratio in Figure 7, the influence of the atmosphere of the heat treatment was small at 300 °C, but the tendency was different depending on the atmosphere of the heat treatment above 300 °C. In other words, it gradually decreased above 300 °C in a nitrogen atmosphere. However, in an oxygen atmosphere, it increased significantly, and its ratio increased to 36% compared to that of the virgin at 500 °C. From this result, it is obvious that in the nitrogen atmosphere, due to the decrease in oxygen functional groups present in the carbon fiber as the temperature of the heat treatment increases, the contact angle increases and the polar surface free energy decreases, and in the oxygen atmosphere, due to the introduction of oxygen functional groups, the contact angle decreases and the polar surface free energy increases.

Other previous studies have reported that when heat-treated to 300 °C in a nitrogen atmosphere, curing of sizing reduces oxygen functional groups and polar free energy [25], and when plasma is treated in an oxygen atmosphere, it increases the carbon fiber surface area and oxygen functional groups [29]. It was also confirmed that after the electrical oxidation treatment, the surface area was increased by etching the carbon fiber surface, and the polar free energy was increased through an increase in C–O, C=O, and O=C–O [7]. From these results, it was confirmed that when surface treatment was performed, the higher the oxygen content in the atmosphere, the more the oxygen functional groups increased, and the polar free energy increased.

Based on the results of the analysis of the mechanical and chemical properties of carbon fibers according to the atmosphere and temperature of the heat treatment, the mechanism of the surface changes and oxygen functional group of carbon fibers are illustrated in Figure 8. During heat treatment in a nitrogen atmosphere, which is an inert gas, as the temperature rose to 1000 °C, the sizing that was present on the surface of the carbon fiber was removed, and the hydroxyl group (C–O) and carbonyl group (C=O) that were present on the surface of the carbon fiber were removed by heat, as shown in Equation (2). From this result, it is evident that the oxygen functional group is gradually reduced compared to the virgin, and it reacts only to the surface of carbon fiber. On the other hand, during heat treatment in an oxygen atmosphere, which is an active gas, sizing is removed below 300 °C and oxygen atoms of the hydroxyl group (C–O) and carbonyl group (C=O) that exist on the surface of carbon fibers and atoms in the oxygen atmosphere are bonded, as shown in Equation (2), and the oxygen functional group decreases. On the other hand, the atoms in the oxygen atmosphere react violently with the carbon existing inside the carbon fiber above 300 °C, and they are removed by CO and CO_2_, as shown in Equation (3). The carbon inside the carbon fiber decreases. Then, atoms in the oxygen atmosphere permeate the surface of the carbon fiber, the hydroxyl group (C–O) and carbonyl group (C=O) increase, and the lactone group (O=C–O) rapidly forms and increases compared with the virgin. From this, it is concluded that the oxygen functionality significantly increases compared to the virgin. However, atoms in the oxygen atmosphere continuously permeate the surface of the carbon fiber and O=C–O increases above 600 °C, but it is judged that the carbon fiber will be lost due to a violent reaction between the atoms in the carbon and oxygen atmosphere inside the carbon fiber.
(2)$$C\text{–}O+O\;\to \;C+O_{2}↑$$
(3)$$\begin{array}{cc} & ↗\;\mathrm{CO}↑+\;O\\ C+O_{2} & \\ & ↘\;\;{\mathrm{CO}}_{2}↑ \end{array}$$

## 4. Conclusions

In this study, we analyzed the mechanical and chemical properties of carbon fibers depending on the temperature of the heat treatment in a nitrogen and oxygen atmosphere and determined the mechanism of the chemical state change and oxygen functional group of the carbon fiber surface depending on the conditions of the heat treatment. In the nitrogen atmosphere, which is an inert gas, the tensile properties were equivalent to those of the virgin up to 500 °C but decreased to 71% with respect to those of the virgin at 1000 °C. As the temperature increases to 1000 °C at room temperature, C–C and C=C on the surface of the carbon fiber increase, and the oxygen atoms of C–O and C=O present in the carbon fiber become removed due to heat. From this, the oxygen functional group was gradually reduced. In addition, the free energy decreased as the temperature of the heat treatment increased in the nitrogen atmosphere. On the other hand, in the oxygen atmosphere, which is an active gas, the tensile properties are not significantly different from those of the virgin up to 300 °C but gradually decrease at 500 °C and deteriorate to an unmeasurable level at 600 °C. Up to 300 °C, most of the C–O and C=O of carbon fibers are removed and the oxygen functional group decreases, but above 300 °C, C–O and C=O increase and O=C–O rapidly forms and increases. The oxygen functionality increased significantly. As the temperature of the heat treatment increased in an oxygen atmosphere, the polar free energy increased.

## Figures and Tables

**Figure 1 polymers-14-02412-f001:**
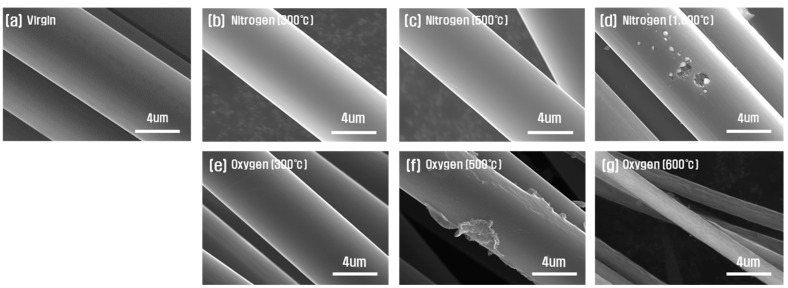
SEM photographs of carbon fibers according to heat treatment conditions: (**a**) Virgin, (**b**) 300 °C, (**c**) 500 °C, (**d**) 1000 °C with nitrogen, and (**e**) 300 °C, (**f**) 500 °C, and (**g**) 600 °C with oxygen.

**Figure 2 polymers-14-02412-f002:**
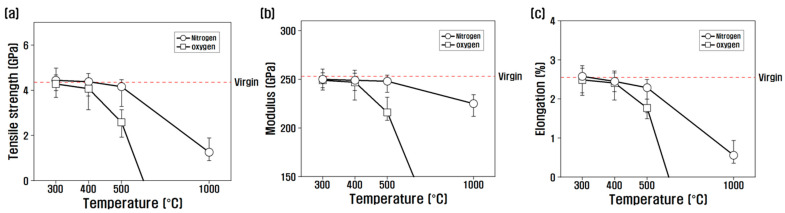
Variation of (**a**) tensile strength, (**b**) modulus, and (**c**) elongation of heat-treated carbon fiber.

**Figure 3 polymers-14-02412-f003:**
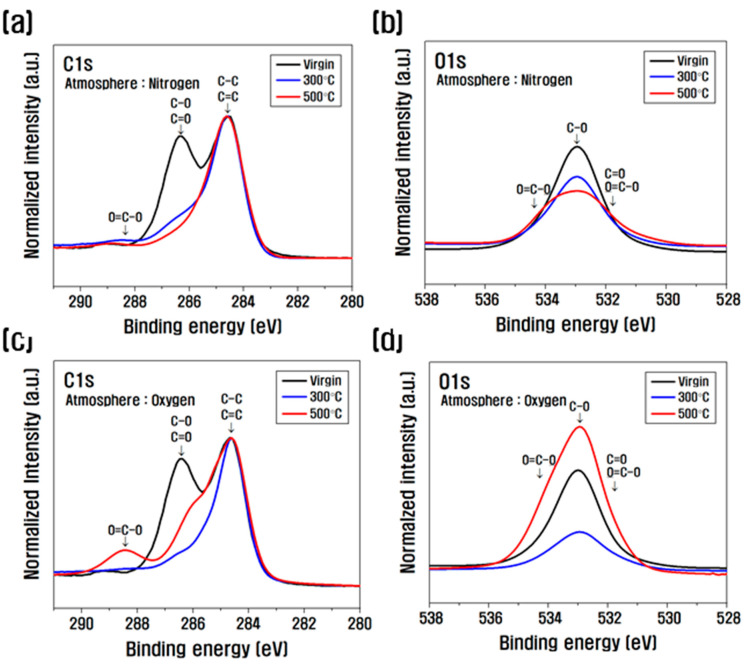
XPS C1s and O1s spectra of heat-treated carbon fibers. (**a**,**b**) Nitrogen atmosphere; (**c**,**d**) oxygen atmosphere.

**Figure 4 polymers-14-02412-f004:**
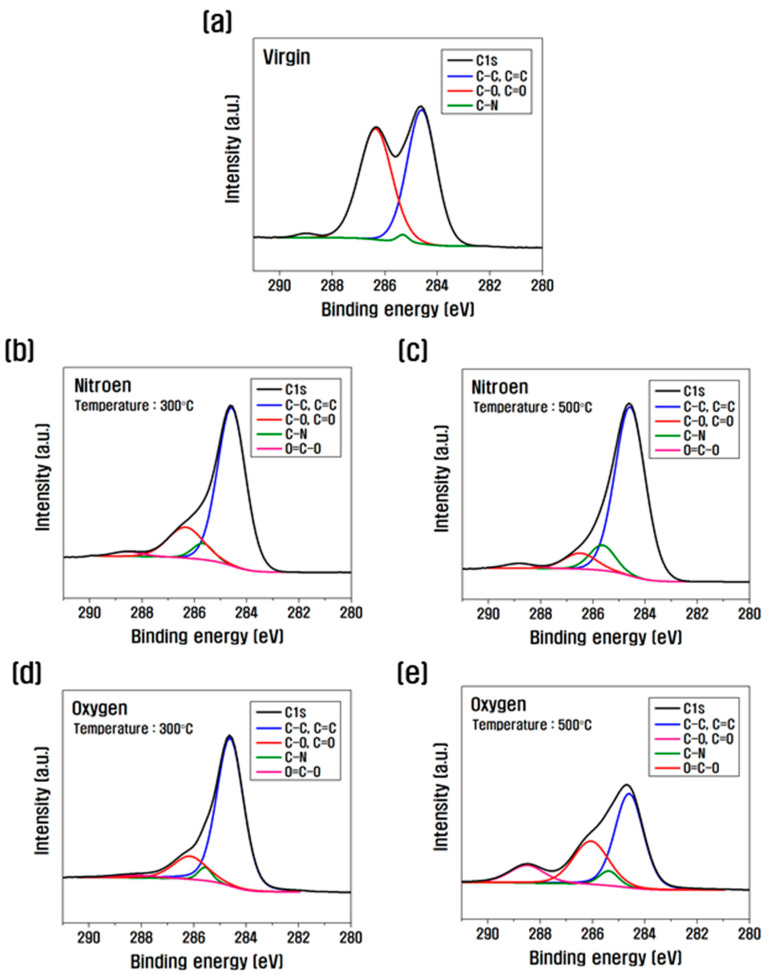
XPS photoelectron spectra of C1s after heat treatment: (**a**) Virgin, (**b**) 300 °C, (**c**) 500 °C with nitrogen, and (**d**) 300 °C and (**e**) 500 °C with oxygen.

**Figure 5 polymers-14-02412-f005:**
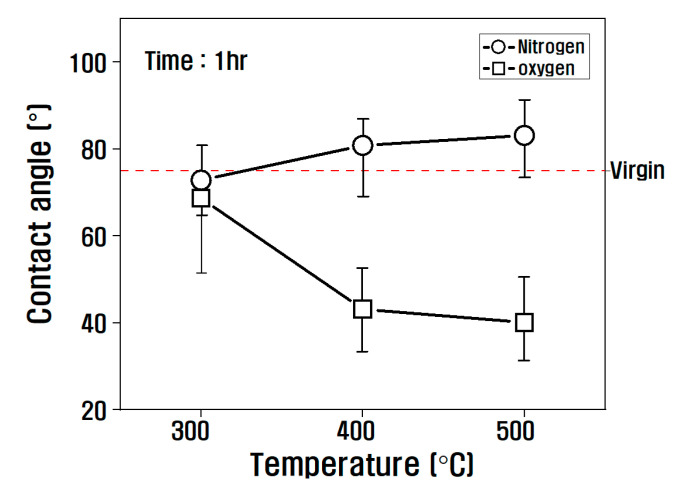
Relationship between contact angle and heat treatment temperature in nitrogen and oxygen.

**Figure 6 polymers-14-02412-f006:**
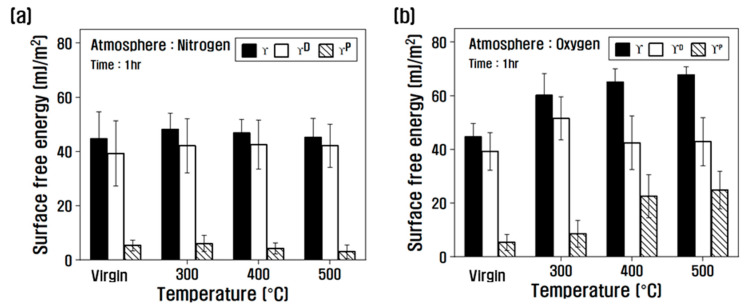
Variation of surface free energy of heat-treated carbon fibers; (**a**) nitrogen atmosphere, (**b**) oxygen atmosphere.

**Figure 7 polymers-14-02412-f007:**
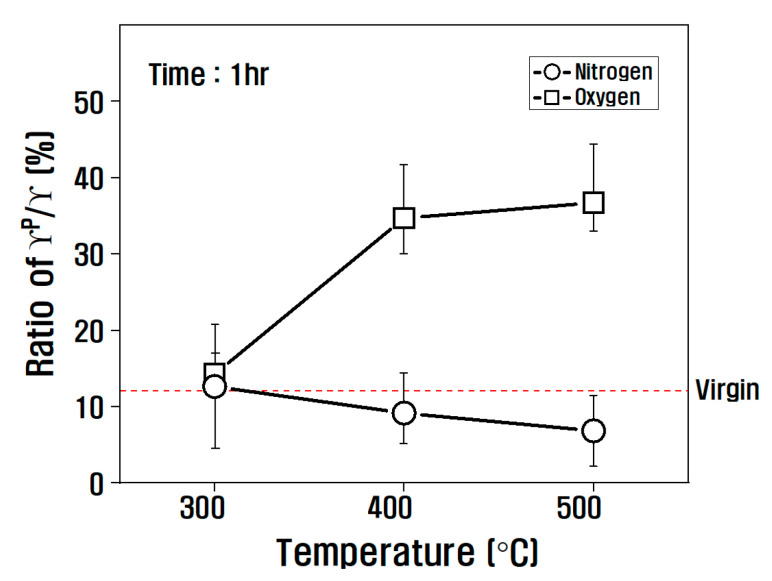
Variation of $\gamma^{P}/\gamma$ according to heat treatment conditions.

**Figure 8 polymers-14-02412-f008:**
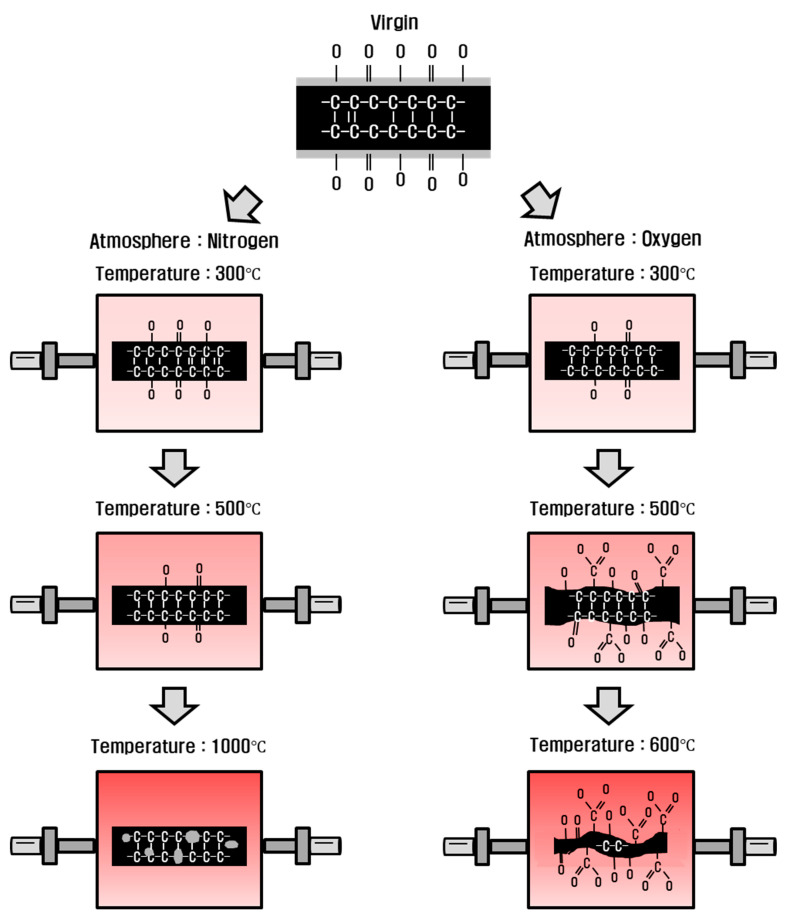
Schematic of the chemical reaction of carbon fiber according to heat treatment atmosphere and temperature.

**Table 1 polymers-14-02412-t001:** Effect of heat treatment condition on the surface elemental composition of carbon fibers.

Treatment Condition		Elemental Composition (at. %)				O/C
Atmosphere	Temperature	Carbon	Oxygen	Nitrogen	Silicon	
Virgin		76.31	21.31	0.75	1.63	0.28
Nitrogen	300 °C	80.15	16.97	1.17	1.71	0.21
	500 °C	85.85	11.13	1.19	1.83	0.14
Oxygen	300 °C	83.61	11.66	2.36	2.37	0.13
	500 °C	68.03	26.50	2.89	2.58	0.39

**Table 2 polymers-14-02412-t002:** Functional group according to atmosphere and temperature by XPS.

Treatment Condition		C1s (at. %)			
Atmosphere	Temperature	C–C, C=C	C–O, C=O	C–N	O=C-O
Virgin		71.09	26.86	0.98	1.07
Nitrogen	300 °C	77.37	19.25	1.46	1.92
	500 °C	85.65	10.05	1.39	2.91
Oxygen	300 °C	83.23	12.05	2.82	1.90
	500 °C	56.80	23.66	4.25	15.29

## Data Availability

Not applicable.

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
