# Peer review of "Effects of Heat Treatment Atmosphere and Temperature on the Properties of Carbon Fibers"

_polymers, 2022, doi:10.3390/polym14122412_

Round 1
Reviewer 1 Report
The carbon fiber used in this study was T700SC manufactured by Toray, Japan, which 88 has an average diameter of about 7 μm and is based on PAN (It is worth explaining what PAN means).
Please explain what equation 1 is about. Everything on the Owens-Wendt approach to determining surface free energy. However, the notation itself is quite strange. There is also no citation to the source manuscript which describes the approach used.
What do the authors mean when they use the term "the polar surface free energy" to mean the component of polar responses?
Reviewer 2 Report
Dear Editor,
Thank you for your invitation for reviewing of polymers-1750356 “Effects of heat treatment atmosphere and temperature on the properties of carbon fibers”
The author was investigated to changes chemical properties on surface of carbon fibers through heat treatment under oxygen or nitrogen. To reveal the effect of heat treatment pf carbon fibers under different atmosphere, they were characterized by FE-SEM (Field Emission Scanning), UTM (Universal Tensile Testers), XPS (X-ray Photoelectron Spectroscopy), mechanical properties and surface-free energy. The result is moderate and the reviewer recommend minor revision.
However, reviewer wonder that the contents is in fields of MDPI polymer or not. I recommend to submit another journal such as fiber, appl. Sci., or Material.
Please consider to the following comments.
1. Would you check volatile components under heat treatment of carbon fiber? If you will identify the gas, we can easily understand the mechanism.
2. They described at line 155. The carbon fiber will be existing number of size and pore through heat treatment. Is it possible to confirm the morphology through Fe-SEM or BET method?
3. Regarding elemental compositions in Table 1, why nitrogen treatment is lower composition than oxygen treatment? And Why silicon is included in it?
4. Regarding % of functional group under treatment in Table 2, Comparison of virgin with treatment one shows increase of O=C-O on the surface. However, the author described removement of CO and CO2 above 300 degrees Celsius at line 299 and eq (3) also. This suggestion is not match as shown in Figure 8.
